Edge computing resource scheduling method based on container elastic scaling

Wang Huaijun
Deng Erhao
Li Junhuai lijunhuai@xaut.edu.cn
Zhang Chenfei
School of Computer Science and Engineering, Xi’an University of Technology , Xi’an, Shaanxi , China
Wang Wei
Electronic publication date: 2024 Oct 11
Publication date: 2024
Volume: 10
Electronic Location ID: e2379
Received 2024 May 23; Accepted 2024 Sep 10
Copyright: © 2024 Wang et al.
Copyright year: 2024
Copyright holder: Wang et al.
License: This is an open access article distributed under the terms of the Creative Commons Attribution License, which permits unrestricted use, distribution, reproduction and adaptation in any medium and for any purpose provided that it is properly attributed. For attribution, the original author(s), title, publication source (PeerJ Computer Science) and either DOI or URL of the article must be cited.
License URL: https://creativecommons.org/licenses/by/4.0/

Keywords: Container elastic scaling, Convolutional neural network, Load prediction, Reinforcement learning

Funding: National Key R&D Program of China 2018YFB1703003 Natural Science Foundation of China 61971347 Key research and development program of Shaanxi Province 2022SF-353 This work was funded by the National Key R&D Program of China (No. 2018YFB1703003), the Natural Science Foundation of China (No. 61971347), and the Key research and development program of Shaanxi Province (2022SF-353). The funders had no role in study design, data collection and analysis, decision to publish, or preparation of the manuscript.

==============================
Edge computing is a crucial technology to solve the problem of computing resources and bandwidth required for extensive edge data processing, as well as for meeting the real-time demands of applications. Container virtualization technology has become the underlying technical basis for edge computing due to its efficient performance. Because the traditional container scaling strategy has issues such as long response times, low resource utilization, and unpredictable container application loads, this article proposes a method for scheduling edge computing resources based on the elastic scaling of containers. Firstly, a container load prediction model (Trend Enhanced-Temporal Convolutional Network, TE-TCN) is designed based on the temporal convolutional neural network, which features an encoder-decoder structure. The encoder extracts potential temporal relationship features from the historical data of the container load, while the decoder identifies the trend item of the container load through the trend enhancement module. Subsequently, the information extracted by the encoder and decoder is fed into the fully connected layer to facilitate container load prediction using the dual-input ResNet method. Secondly, Markov decision process (MDP) is used to model the elastic expansion problem of containers in multi-objective optimization. Utilizing the prediction outcomes of the TE-TCN load prediction model, a time-varying action space is formulated to address the issue of excessive action space in conventional reinforcement learning. Subsequently, a predictive container scaling strategy based on reinforcement learning is devised to align with the application load patterns in the container environment, enabling adaptation to the surge in traffic generated by the container environment. Finally, the experimental results on the WorldCup98 dataset and the real dataset show that the TE-TCN model can accurately predict the container load change. Experiments in the actual environment demonstrate that the proposed strategy reduces the average response time by 16.2% when the burst load arrives, and increases the average CPU utilization by 44.6% when the jitter load occurs.

Introduction

The vigorous development of the Internet of Things is inseparable from the support of edge computing technology. Data fusion, data analysis, network security, and data security at the edge of the network all need to rely on edge computing technology (Ahmed et al., 2024). At the same time, in the face of the growing data transmission and computing needs of the Internet of Things, edge computing can effectively cope with the lack of computing power of the Internet of Things equipment itself. It can also fully alleviate network congestion and other issues (Jeremiah, Yang & Park, 2024). For instance, in real industrial applications, real-time control of the production workshop site is achieved by installing an edge gateway within the workshop. At the same time, it can also establish an industrial control platform through edge cloud, and use the data analysis and decision-making ability provided by the platform to provide a decision-making basis for real-time control (Ahmadi, 2024). In many industries, such as industrial control, most businesses are deployed on the platform using containerization. Container technology offers a high level of flexibility, allowing for rapid expansion and contraction to accommodate varying load requirements and meet the service-level agreement (SLA) of enterprises. However, the current resource scheduling capabilities of mainstream operating systems are limited, leading to uneven resource utilization. This issue is particularly evident in scenarios with significant fluctuations in application load, resulting in either resource wastage or inadequate resources. Therefore, through the elastic expansion of the container, the containerized application can dynamically adjust the number of container instances of the service according to the elastic expansion strategy and application status (Nawrocki, Osypanka & Posluszny, 2023). This helps improve resource utilization and ensures reasonable resource scheduling.

The Horizontal Pod Autoscaler (HPA) and Vertical Pod Autoscaler (VPA) are container elastic scaling strategies. In the Horizontal Pod Autoscaler, the two most common optimization objectives are to reduce the rate of SLA violations and improve CPU utilization (Karol Santos Nunes et al., 2024; Daradkeh & Agarwal, 2023). According to the trigger conditions and execution methods, there are two common ways to implement the container elastic scaling strategy: (1) adjusting thresholds based on the load prediction model; and (2) using a reinforcement learning agent to make scaling decisions and iteratively learn the best scaling strategy (Khaleq & Ra, 2021; Zhong et al., 2022; Huo et al., 2022; Ahmad et al., 2022). Currently, these two technologies have garnered significant attention in academia. Traditional time series prediction methods utilize statistical prediction models known for their low prediction cost, simplicity, and speed, such as moving average (MA) and autoregressive (AR) (Bauer et al., 2020). However, these methods typically only consider the time series data as input for the model, overlooking potential temporal relationships and trend characteristics. This limitation hinders the traditional models from accurately and efficiently predicting container loads. the most commonly used reinforcement learning algorithm for container elastic scaling strategy is the Q-Learning algorithm (Agarwal, Rodriguez & Buyya, 2021). However, because the state space and action space in the container environment are continuous, when the state or action space is very large, a large amount of calculation will be generated, which is difficult to cope with the sudden traffic generated in the container environment. At the same time, in the container scheduling environment, when the state of the application changes quickly, the traditional Q-Learning scaling strategy has the problem of simplification of scaling conditions and untimely scaling, so the trained reinforcement learning agent performs poorly.

In light of the aforementioned issues, this article introduces a method for scheduling edge computing resources based on container elastic scaling. It utilizes reinforcement learning to develop a container scaling strategy with dual optimization goals: reducing SLA violation rates and enhancing resource utilization. By leveraging the time convolutional neural network to extract potential temporal relationship features from historical container load data and incorporating load trend features from the trend enhancement module, the TE-TCN container load prediction model is devised to offer more precise container load predictions. Then MDP is used to model the container elastic scaling problem in multi-objective optimization. The fixed-size action space in traditional reinforcement learning is transformed into a time-varying space combined with the TE-TCN model load prediction value. A predictive container scaling strategy based on reinforcement learning is devised to adjust to the burst traffic generated by the container environment, decrease the response time of the application, and enhance the utilization efficiency of edge computing cloud resources. Following experimental testing on the WorldCup98 dataset and a real dataset, the proposed method can accurately predict the container load changes, and the container application performs effectively amidst load fluctuations in the actual environment.

The main contributions of this work include:

(1) Based on the container scaling strategy, we propose a resource scheduling method for edge computing based on container elastic scaling.

(2) The TE-TCN container load prediction model is designed to combine the potential temporal relationship characteristics in historical data with the load trend characteristics in the trend enhancement module, in order to predict the future container load more accurately.

(3) A predictive container elastic scaling strategy based on an enhanced Q-Learning algorithm is designed. This strategy enables faster expansion action when a burst load arrives, reduces the response time of the application, and ensures higher CPU utilization during idle periods to avoid resource wastage.

The rest of this article is organized as follows. ‘Related research’ introduces the related research of cloud platform load prediction and container elastic scaling. In ‘TE-TCN container load prediction model’, the specific structure of the TE-TCN container load prediction model is introduced, and the mathematical formulas of related problems are provided. ‘Predictive container scaling strategy based on reinforcement learning’ introduces the predictive container elastic scaling strategy based on reinforcement learning and the edge computing resource scheduling model. The experiments of the method proposed in this article are shown in ‘Experiments’. Finally, a conclusion is drawn in ‘Conclusion’.

Related research

Cloud platform load prediction

The original prediction method utilizes traditional statistical prediction models, which are characterized by low cost, simplicity, and fast prediction. These models include the MA and AR model, etc. Jiang et al. (2013) initially demonstrated that the load characteristics of data centers can be effectively represented by time series statistical models. Since then, researchers have utilized various models to study them. Jananee & Nimala (2023) predicted future workload using an autoregression and moving average (ARMA) model and proposed a comprehensive cloud resource allocation scheme that combines prediction and automatic expansion technology. Mishra et al. (2023) used autoregressive integrated moving average (ARIMA) to predict cloud workloads, demonstrating its high accuracy and stability in predicting workload performance. Mazumdar & Kumar (2018) used wavelet decomposition and Kalman filtering to enhance the ARMA model, thereby improving its predictive capability and efficiency. Jiang, E & Song (2018) introduced an online feedback mechanism based on AR and continuously updated the AR model by monitoring the cloud load online.

With the rapid growth of data size, traditional statistical methods have been unable to effectively deal with the evolving characteristics of large-scale data. Deep learning provides a new approach for cloud load prediction. Zhang et al. (2016) used recurrent neural networks (RNN) to achieve high-precision cloud workload prediction, but studies have shown that it has limited performance under long-term sequence prediction tasks. Song et al. (2018) used a long short-term memory (LSTM) network to predict future cloud load. Experiments on Google Cloud load and distributed load data sets proved that LSTM can effectively learn long-term dependencies in cloud load. Cao et al. (2017) applied the concept of time convolution to predict data center traffic and proposed a model called ITRCN. This model treats traffic data as one-dimensional grid data on the time axis for learning and demonstrates superior load prediction performance compared to CNN. Sutskever, Vinyals & Le (2014) initially proposed a sequence-to-sequence (Seq2Seq) method for learning time series data and proved that this structure can achieve high performance. Dogani, Khunjush & Seydali (2023) enhanced the accuracy of host load prediction in cloud environments by combining bidirectional gated recurrent units (BiGRU), discrete wavelet transform (DWT), and attention mechanisms. Subsequently, Azizi et al. (2024) summarized and structured their research to tackle service placement and load distribution challenges in Internet of Things (IoT)-based fog computing networks. Predić et al. (2024) employed a decomposition-assisted attention recurrent neural network, conducting hyperparameter optimization through an enhanced particle swarm optimization algorithm to forecast the load of cloud computing resources. Experimental results illustrated that their model exhibits strong predictive capability and adaptability. Therefore, the prediction accuracy of deep learning method is better than that of traditional statistical prediction method in cloud resource load prediction. However, the prediction accuracy of deep learning still has room for improvement.

Container elastic scaling

Khatua, Manna & Mukherjee (2014) predicted the application load using ARIMA and implemented scaling actions before the requests arrived to enhance resource utilization. However, the ARIMA prediction model has a relatively slow prediction speed, and it does not yield a practical application effect when confronted with the rapidly changing workload in the container environment. Iqbal, Erradi & Mahmood (2018) used deep learning methods to predict workloads in the next time interval, and the prediction results were used to automatically extend Web applications. Zhao et al. (2019) used a combination of the empirical mode decomposition method and the ARIMA model to predict the load of the Pod and adjust the number of Pods before the predicted peak reaches, so as to solve the problem of response delay during container expansion. Toka et al. (2021) proposed a Kubernetes extension framework, in which various machine learning prediction methods compete with each other to provide the most suitable scaling strategy for the actual situation. Experiments show that the framework can significantly reduce the number of lost requests. The scaling strategy based on dynamic thresholds can make scaling decisions in advance. The threshold setting process is often considered too complicated, and determining the number of instances to expand or shrink after reaching the threshold can be difficult.

The scaling strategy based on reinforcement learning is a predictive strategy, which was initially used to address server or virtual machine scheduling problems (Sheng et al., 2022). However, the reinforcement learning method that performs well in the virtual machine scheduling scenario cannot be directly applied to the container elastic scaling scenario. In the container scenario, Horovitz & Arian (2018) proposed a method that uses reinforcement learning to dynamically adjust the scaling threshold of application instances. The state is determined based on the number of instances, leading to a reduction in the state space size of the traditional reinforcement method. However, this method still belongs to the threshold-based method. Rossi, Nardelli & Cardellini (2019) proposed a model-based reinforcement learning method, which uses the system model to address the hybrid scaling strategy problem in the container scenario. However, the fixed number of actions corresponding to each state ignores the rapidly changing workload in the container environment, leading to more SLA violations or resource wastage. Therefore, the state and action space size of reinforcement learning in container elastic scaling scene still need to be optimized.

Te-tcn container load prediction model

Aiming to address the challenge of predicting the complex and variable container application load using traditional time series prediction models, a container load prediction model (TE-TCN) is designed based on the temporal convolutional neural network, as illustrated in Fig. 1.

Figure 1 Architecture of TE-TCN.

The model adopts the sequence-to-sequence architecture. Based on the encoder, the potential temporal relationship features are extracted from the historical data of the container load to help the model learn the long-term changes of the container load, in order to predict the future container load more accurately. Additionally, in the trend enhancement module, local weighted regression is utilized to extract the trend term of the load history as the input of the decoder to enhance the trend characteristics of the sequence. Subsequently, the representation of the historical data output by the encoder is acquired, and the historical representation of the encoder output is merged with the future trend representation extracted by the trend enhancement module using the dual-input ResNet method. Finally, the output of the dual-input ResNet goes through the fully connected layer to achieve high-precision container load prediction. The TE-TCN model exhibits translation invariance and is insensitive to the position of the input sequence. Consequently, for the translated sequence, the prediction result should resemble the original sequence, making it more suitable for handling container load prediction tasks.

Load prediction problem modeling

Suppose a set of historical load data {yt−k,...,yt−1,yt} for container applications is stored in the container cluster database. Here, yt represents the total amount of web requests for the cloud application during period {t;t∈T}, and t represents the time subscript for the web requests.

The container load prediction model retrieves historical load data from the database and uses the extracted trend item information {xt−k,...,xt−1,xt} of the container load as input for the model. According to historical container load data {yt−k,...,yt−1, yt} and trend item {xt−k,...,xt−1,xt}, the prediction for the number of web requests {y^t+1,y^t+2,...,y^t+h} in the next n time steps is made.

The time series prediction strategy can typically be categorized into single-step prediction and multi-step prediction based on the number of predicted future values. Common multi-step prediction strategies include the iterative multi-step prediction method and the direct multi-step prediction method.

Using a single-step prediction method, the prediction problem of predicting the amount of web requests y^t+1 at a future time step can be described by the following Formula (1):

(1) yt+1∧=f(xt−k,...,xt−1,xt,yt−k,...,yt−1,yt)

Among them, y^t+1 represents the predicted value of the future t+1 step cloud application web request amount at time step {t;t∈T}. The function f(⋅) represents the prediction function of the model, and k is the length of the historical data considered.

Using the iterative multi-step prediction method, the prediction problem of predicting the number of web requests {yt+1,yt+2, ...,yt+h} in the future h time steps can be described by Formula (2):

(2) y^t+1=f(xt−k,...,xt−1,xt,yt−k,...,yt−1,yt)y^t+2=f(xt−k+1,...,xt−1,xt,xt+1,yt−k+1,...,yt−1,yt,yt+1)⋮y^t+h=f(xt−k+h−1,...,xt+h−2,xt+h−1,yt−k+h−1,...,yt+h−2,yt+h−1).

Among them, {y^t+1,y^t+2,...,y^t+h} represents the predicted value of the future amount of {t+i;t∈T,i∈(1,2,...,h)} step cloud application web requests at time step {t+i;t∈T,i∈(0,1,...,h−1)}, respectively.

Using the direct multi-step prediction method, the prediction problem of predicting the amount of web requests {yt+1,yt+2, ...,yt+h} in the future h time steps can be described by Formula (3):

(3) y^t+h|t=f(xt−k,...,xt−1,xt,yt−k,...,yt−1,yt,h)

Among them, y^t+h|t represents the predicted value of the web request amount of the cloud application in the future t+h step at time step {t;t∈T}, yt represents the sampling and sum of the web request amount of the cloud application in the period t, xt is the trend term at the t time step, and h represents the period to be predicted.

Encoder constructed by multi-layer hole causal convolution

The encoder comprises multiple layers of dilated causal convolutions, with each layer extracting abstract features across different cycles. It can effectively extract the long-term and short-term relationship characteristics from the data. The encoder’s function is to transform the input sequence into an intermediate vector that represents the key features and patterns of historical data.

When predicting container load for univariate sequences, the encoder can learn the mapping from input sequence X=(xt−k,...,xt−1,xt) of length k+1 to X to ht, as shown in Formula (4):

(4) ht=f(ht−1,xt).

Among them, ht represents the hidden state of the encoder at time t, and f(⋅) represents the multi-layer hole causal convolution.

The input sequence is X=(xt−k,...,xt−1,xt), and the convolution kernel size is ω. The feature vector output at time t is as follows Formula (5):

(5) s(t)=∑k=0K−1ω(k)x(t−d⋅k).

Among them, d represents the expansion factor, and K represents the size of the convolution kernel.

To prevent the issue of gradient disappearance, multiple convolutional layers are connected in series using residual connections to create a multi-layer hollow causal convolution. Multi-layer dilated causal convolution can expand the receptive field without increasing the number of parameters. This allows for capturing long-range time dependence and improving the effectiveness of feature extraction.

Fusion trend enhancement module decoder

In the decoder section, the encoder’s output representing historical data is initially acquired. Subsequently, the trend enhancement module utilizes the trend term extracted through local weighted regression as input to amplify the trend characteristics of the sequence. Lastly, the dual-input ResNet technique is employed to merge the historical representation with the future trend representation, resulting in output results of increased accuracy through the fully connected layer. The function of the decoder is to decode the intermediate vector representing the key features and laws of historical data and generate the load prediction results. The structure of the decoder model is shown in Fig. 2.

Figure 2 Architecture of decoder model.

The trend enhancement module in the decoder enables the network model to fully consider the evolving trend information in the feature factors over time. In this module, the Seasonal and Trend decomposition using Loess (STL) method is chosen to extract the long-term trend of cloud application load data.

In container load forecasting, the time series decomposition method can decompose the container load sequence data into multiple components. It is helpful to understand and predict the trend of container load sequence data. Specifically, the time series data can be divided into trend items, seasonal items and residual items. The load data of cloud applications can be expressed as a function of three factors, namely yt=f{Tt,St,Rt}. According to the characteristics of time series, the additive model and multiplicative model are chosen for decomposing the time series data. In the additive model, each moment of the time series is calculated by adding the trend term, the seasonal term, and the residual term. The addition model for processing time series data can be described as Formula (6):

(6) yt=Tt+St+Rt.

Among them, Tt represents the trend value at time t, St represents the seasonal value at time t, Rt represents the residual value at time t, and yt represents the value of the time series at time t.

In the multiplicative model, the value of each time point in the time series is calculated by multiplying the trend term, the seasonal term, and the residual term, as described by Formula (7):

(7) yt=Tt×St×Rt.

The STL method uses the additive model to decompose the HTTP requests in the WorldCup98 dataset. The WorldCup98 dataset experiences significant changes over the weekend, showing strong seasonality. In this article, the STL method with a decomposition period of 1 day is chosen to extract the trend term. Given historical data xt with a length of n, the trend term Tt is obtained through local weighted regression estimation, indicating the long-term trend change of the sequence. Specifically, the local time series window is initially shifted, and a polynomial is applied to fit the data within each local window to obtain the trend term of the data within the window. The results of trend enhancement can be described by Formula (8):

(8) TE(xt)=Tt+xt=Loess(t,{xt}t=1n)+xt.

Among them, Tt represents the sequence trend term, which signifies the change in the long-term trend of the sequence. Loess stands for locally weighted regression, which is equivalent to fitting the trend term of the sequence within a local window.

Predictive container scaling strategy based on reinforcement learning

Aiming to address the current challenges of responsive and predictive container elastic scaling strategies, this article proposes a predictive container elastic scaling strategy based on reinforcement learning (PRL). Firstly, the issue of elastic expansion and contraction of containers is modeled using MDP. According to the prediction results of the TE-TCN container load predictio0n model in the previous section, a time-varying workspace has been designed to address the issue of excessive action space in traditional reinforcement learning. Secondly, addressing the complex nature of application load in a container environment, an enhanced Q-Learning algorithm is developed to facilitate predictive container elastic scaling. Next, the architecture of the predictive container elastic scaling model is introduced.

Markov decision process of container elastic scaling

MDP is a discrete-time stochastic control process that serves as a mathematical framework for modeling decision problems. MDP describes the process of an agent making decisions in an environment. The environment comprises a set of states and a transition probability distribution. The agent selects an action based on the current state and possible actions, and updates the strategy according to the feedback from the environment. MDP has a Markov property when it is used in the system environment (it is conditionally independent of the past state when given the current state) to simulate the random strategy implemented by the agent and the rewards it brings. The Markov decision process (MDP) is used to model the container elasticity problem as a 4-tuple {S, A, R, P}, where S represents the state set, A represents the action set, P represents the state transition probability, and R represents the reward. Each part is defined as follows:

(1) Discrete state space

The state set defined by each state st∈S indicates that the state of the application at time i consists of two components: the CPU utilization ui at time i and the response time ri at time i. To discretize the state space, this section divides the CPU utilization into m levels and the response time into n levels. The discretized CPU utilization is represented by ui∈{0,u⋅,2u⋅,...,mu⋅}, and the response time is represented by ri∈{r⋅,2r⋅,...,nr⋅}, where ui and ri are the smallest units for discretization.

(2) Time-varying action spaces

The time-varying action spaces consist of three actions: A=(−1,0,+1), where +1 indicates the expansion action Aup, −1 indicates the contraction action Adown, and 0 indicates that no action is performed. The specific size of the scaling is determined by the CPU utilization prediction value and the SLA violation rate.

Specifically, the expansion action represents the addition of ⌈ρ%×I⌉ container instances, where ρ% represents the current SLA violation rate, and I represents the current number of container instances. The scaling action represents the reduction of ⌊0.5×(c−p)⌋ container instances, where c represents the currently allocated CPU, and p represents the predicted value of CPU utilization in the next cycle. Through three types of actions and dynamic action values, it is possible to quickly respond to a workload that changes drastically in a short time and solve the problem of a fixed action space.

(3) Reward function

R(st,at) represents the anticipated reward for choosing action at in state st. To prevent SLA violations and maximize CPU utilization, it is essential to develop a reward function that considers response time, the SLA-specified response time threshold (RTTH), and CPU utilization. The reward function formulated according to the above principle can be expressed as Formula (9):

(9) R={1−e−P(1−respTimeRTTH)1−ρ,respTime>RTTH1−e−p1−ρ,other.

Among them, respTime represents the response time, RTTH represents the response time threshold specified by the SLA, with RTTH set to 146 ms, see Section ‘Container elastic expansion experiment’ for detailed reasons, ρ represents CPU utilization, and P is a manually set constant.

Container elastic scaling Q-Learning algorithm

In a container scheduling environment, the application’s state changes rapidly. It is necessary to collect metrics in the current cycle to obtain the current state s, rather than directly observing the state s based on the state after the operation was performed in the previous cycle. At the beginning of each cycle, the container elastic scaling Q-Learning algorithm first retrieves the access traffic, CPU utilization, and response time of the container cloud application from the monitoring module, and identifies the current state s of the container environment. The load prediction module predicts the container CPU utilization in the next cycle based on the performance data of the current cycle to help the scaling module adapt to the ever-changing workload in the container environment. The container scaling module changes the scaling action in the space when using the ε−greedy strategy selection and obtains the new state st+1 and reward r, and optimizes the scaling strategy by updating the Q table. The pseudo-code of Q-Learning algorithm in container scaling is shown in Algorithm 1.

Algorithm 1 Q-Learning algorithm in container scaling.

Require: The current container state s	
Ensure: The container expansion action a	
 1: Initialize Q (s, a)	
 2: Observing container state s	
 3: Choose action a according to Q (s, a)	
 4: Cycle until it stops	
 5: Observe the current SLA violation rate ρ, the total amount of CPU currently allocated c, and the predicted value p of CPU utilization in the next cycle.	
 6: Use ε−greedy to select action a, and dynamically adjust the number of container instances according to indicators ρ, p, and c.	
 7: Get reward r and new state st+1.	
 8: According to Formula (10), update Q (s, a).	
 9: When the termination state is reached, the next round begins.	

After initializing the Q table, the algorithm performs the following steps in each iteration until convergence.

(1) Observation: The monitoring module retrieves the current state s of the application, while the prediction module computes the anticipated CPU utilization value p for the next cycle based on the gathered metrics.

(2) Selection: The allocation module uses ε−greedy strategy to select the scaling action a with the largest profit in the current state.

(3) Executing: The executing module executes action a and dynamically adjusts the number of container instances based on indicators ρ, p, and c.

(4) Feedback: The agent obtains the reward r and state st+1 brought by the action a, and updates the Q table through the Formula (10):

(10) Qt+1(s,a)←Qt(s,a)+α⋅[r+γ⋅maxa′∈A(s′)⁡Qt(s′,a′)−Qt(s,a)].

Among them, t represents the number of iterations, s represents the current state, a represents the action taken by the agent in the current state, r represents the reward obtained, γ∈(0,1) represents the discount factor, α represents the learning rate 0<α≤1, s′ represents the next state, and a′ represents the next action.

Edge computing resource scheduling model based on container elastic scaling

According to the method proposed above, the edge computing resource scheduling model based on container elastic scaling generally consists of four processes: observation, prediction, allocation, and execution, as shown in Fig. 3.

Figure 3 Architecture of predictive container elastic scaling model.

The observation is carried out by the monitoring module, and all traffic accessing the cloud application is directed to the gateway. The gateway evenly distributes traffic to each application instance Pod of the cloud application, and the Envoy component of each Pod forwards the traffic to the Mixer component. Through the Prometheus monitoring component, metrics collected from the Mixer can be obtained and stored. The TE-TCN load prediction module implements the prediction by taking the historical CPU utilization of the application as input and producing the predicted CPU utilization value for the next cycle in the application. The allocation is carried out using the Q-Learning algorithm, which is part of the container scaling strategy. The executor is primarily responsible for interacting with the container, utilizing the API Server to implement the resulting container scaling strategy.

Experiments

In order to comprehensively verify the effectiveness of this method, the experiment is divided into two parts. The first part involves verifying the effectiveness of the TE-TCN container load prediction model. The second part focuses on validating the predictive container scaling strategy based on reinforcement learning under specific load pressure testing conditions.

Experimental analysis of TE-TCN container load prediction model

Dataset

(1) WorldCup98 dataset

This dataset has been widely used in cloud computing-related literature to evaluate elastic scaling strategies (Yang, Pan & Liu, 2024; Bali et al., 2024; Tran & Kim, 2024; da Silva et al., 2022). In this article, the data size is adjusted appropriately according to the preprocessing method in Imdoukh, Ahmad & Alfailakawi (2020). This accelerates the training process of the prediction model and enhances the generalization ability of the model. The workload HTTP request in the WorldCup98 dataset within the same second but different milliseconds is converted into a total workload HTTP request per second. Similarly, the workload HTTP request within the same second but different minutes is converted into the maximum workload HTTP request per minute. The preprocessed dataset represents the maximum load per minute, with approximately 70% of the dataset used as the training set M1, and about 30% of the dataset used as the test set M2, as illustrated in Table 1.

Table 1 The preprocessed WorldCup98 dataset.

Dataset	Description	Size	
The preprocessed WorldCup98 dataset	Training sets M1	87,710	
	Testing sets M2	37,590	

(2) Real load dataset

The application’s efficiency is reflected in various resource metrics, including CPU, memory, disk, and request rate. For the computationally intensive containerized applications used in this experiment, the CPU is the most influential resource. Therefore, this experiment selects CPU utilization as an intuitive indicator of the actual load on the application. In the experiment, the php-apache application was tested in a Kubernetes cluster environment. The Prometheus monitoring tool was used to collect real-time performance indicators of the application, and a 30-min dataset of real load data was generated. The data acquisition interval is 5 s, and the total amount of data is 360. The first 75% of the dataset is used to train the model, and the remaining data contains a burst workload to quantitatively evaluate the prediction effect of each model.

Evaluating indicator

In the load data prediction experiment, four common evaluation indexes are utilized to analyze and evaluate the container load prediction effect. Mean square error (MSE) and R-Square reflect the prediction ability and model fitting of the prediction model, respectively. Additionally, mean absolute error (MAE) and root mean squared error (RMSE) are also taken into consideration.

The MSE describes the average value of the square of the difference between the model’s predicted result and the actual result. Generally, a smaller value is preferable. The calculation method for MSE can be described as follows Formula (11):

(11) MSE=1n∑i=1n(yi−y¯i)2.

The R2 score describes the mean square error of the decrease relative to the variance of the total data. Generally, the closer the score is to 1, the better. The R2 calculation method can be described as follows Formula (12):

(12) R2=1−∑i=1n(yi−y^i)2∑i=1n(yi−y¯i)2.

The MAE describes the average value of the absolute difference between the model’s prediction result and the actual result. Generally, the smaller the error, the better. The calculation method for MAE can be described as Formula (13):

(13) MSE=1n∑i=1n|(yi−y¯i)|.

The RMSE is equal to the square root of the squared difference between the predicted value and the actual value. Generally, the closer it is to 0, the better. The RMSE calculation method can be described as Formula (14):

(14) RMSE=1n∑i=1n(yi−y¯i)2.

In the calculation formula of the above evaluation index, y represents the real value of the time series, y^ represents the predicted value, y¯ represents the average value of the real value, and n represents the length of the time series.

Encoder structure experiment

To establish the reasonable basic parameters of the TE-TCN model, three encoder model structures were implemented: a four-layer structure for void factor d={1,2,4,8}, a five-layer structure for void factor d={1,2,4,8,16}, and a six-layer structure for void factor d={1,2,4,8,16,32}. The results of the validation set val _loss when the model runs 200 epochs under the three encoder structures are shown in Fig. 4.

Figure 4 Val _Loss over 200 epochs of the model with three different encoder structures.

It can be observed from the figure above that the five-layer and six-layer encoder structures outperform the four-layer structure. This is mainly due to the relatively shallow nature of the four layers, and the observation of historical data is not comprehensive. The difference between the five-layer and six-layer structures is minimal, indicating that once the number of hidden layers in the encoder structure reaches a certain threshold, the impact of adding more layers becomes negligible. Therefore, the fundamental parameters of the TE-TCN model in the following experiments in this section have been established, as indicated in Table 2.

Table 2 TE-TCN model parameters.

Parameter	Content	
Input size	180	
Output size	1	
Expansion factors	1, 2, 4, 8, 16	
Kernel size	3	
Activation function	ReLU	
Loss function	MSE	
Optimizer	Adam	
Learning rate	0.001	

Container load prediction experiment

After determining the appropriate model structure, this section aims to comprehensively evaluate the performance of the container load prediction model proposed for burst load arrival. The comparison includes the prediction model TE-TCN with LSTM, ARIMA, and ANN prediction models on the WorldCup98 dataset and a real dataset.

(1) WorldCup98 dataset load prediction

In the experiment, to compare the predictive effects of LSTM, ARIMA, ANN, and TE-TCN models, the M1 dataset was utilized for training the models. The predictive performances of each model were then compared using evaluation indices on the M2 dataset. The single-step prediction results are presented in Table 3. It is evident that, apart from the subpar predictive performance of the ANN model, LSTM, ARIMA, and TE-TCN exhibit comparable performance in terms of prediction accuracy and interpretability.

Table 3 Comparison of single-step prediction results.

Model	R2	MAE	MSE	RMSE	
ANN	0.9360	0.0353	0.0017	0.0412	
ARIMA	0.9864	0.0136	0.0003	0.0177	
LSTM	0.9810	0.0141	0.0005	0.0224	
TE-TCN	0.9853	0.0138	0.0002	0.0141	

Then, the performance change of the model in the case of the growth of the prediction sequence step size is explored. Due to the issue of error escalation with the prediction step size in the recursive prediction strategy, the prediction model adopts a direct prediction strategy with a step size of five. The forecast results are presented in Table 4. It is evident that the TE-TCN model can uphold high prediction accuracy even as the step size of the prediction sequence increases.

Table 4 Comparison of mutli-step (5 steps) prediction results.

Model	R2	MAE	MSE	RMSE	
ANN	0.9320	0.0349	0.0018	0.0424	
ARIMA	0.9291	0.0363	0.0018	0.0424	
LSTM	0.9270	0.0410	0.0020	0.0447	
TE-TCN	0.9566	0.0257	0.0011	0.0312	

Figure 5 illustrates the trend of MSE values for different models as the step size of the prediction sequence increases. By observing the change in the MSE value curve with the increase in prediction step size, it can be seen that the MSE value of the ANN model is 0.0012 to 0.0015 higher than that of other models when the prediction step size is 1. The other three models have similar MSE values. Specifically, the ANN model exhibits poor prediction accuracy, while the other models demonstrate comparable prediction accuracy. As the prediction step size increases, the TE-TCN model demonstrates a gradual improvement in prediction accuracy. When the prediction step reaches five, the prediction accuracy of the LSTM, ANN, and ARIMA models is similar. However, the prediction speed of LSTM and ANN models is faster than that of ARIMA. The TE-TCN model has an MSE value about half as low as the other three models.

Figure 5 The changing trend of MSE values of each model with prediction horizons of 1 and 5 steps.

(2) Real load dataset prediction

In order to clarify the predictive performance of each model using real load data, the prediction accuracy of each model is compared using four evaluation indices, as shown in Table 5. The optimal result data has been highlighted in the table. It is evident that the prediction error of the TE-TCN model is the smallest, and the accuracy is the highest. This demonstrates that the TE-TCN prediction model yields the best predictive performance when faced with sudden load changes in the container environment. Consequently, it can offer a solid data foundation for container resource scheduling.

Table 5 Results of real workload dataset.

Model	R2	MAE	MSE	RMSE	
ANN	0.90514	0.00992	0.00018	0.01360	
ARIMA	0.91794	0.00903	0.00016	0.01264	
LSTM	0.92551	0.00796	0.00014	0.01205	
TE-TCN	0.94495	0.00698	0.00012	0.01089	

In order to compare the load prediction effect of the model in the real container environment, the first 75% of the real dataset is used to train the model, and the load prediction experiment is performed on the last 25% of the dataset. The prediction results of TE-TCN and LSTM prediction models are shown in Fig. 6. Clearly, when the burst load of the application in the container environment occurs, the predicted value of the TE-TCN prediction model is closer to the true value, providing a more reliable basis for decision-making in the predictive container elastic scaling strategy.

Figure 6 Model prediction results under real workload dataset.

Experimental analysis of predictive container scaling strategy based on reinforcement learning

Experimental environment

In order to conduct the container elastic scaling experiment, a Kubernetes cluster is established. The Kubernetes cluster comprises one master and three nodes. This section presents the configuration details of the Kubernetes cluster in Table 6, and the primary software environment information utilized by the Kubernetes cluster in Table 7.

Table 6 Kubernetes cluster configuration.

Virtual machine name	Node type	Number of CPU cores (units)	Memory size (GB)	
Master	Master node	4	16	
Node1	Work node	2	16	
Node2	Work node	2	16	
Node3	Work node	2	16	

Table 7 Kubernetes cluster software environment.

Software name	Description	
Promethus	Provide resource monitoring indicators	
Kubectl	Manage Kubernetes cluster	
Istio	Tools to provide secure connectivity	
Grafana	Visualization of indicators	
Flannel	Network connectivity tool between nodes	
Kube-apiserver	Kubernetes cluster call interface	
Ingress-Nginx	Open services to clients outside the cluster	

Container elastic expansion experiment

In this experiment, two container elastic scaling strategies are compared: the HPA in Kubernetes and the PRL proposed in this article. To compare the performance of the two container elastic scaling strategies in the same environment, the load test application PHP-apache is first deployed in the Kubernetes cluster experimental environment. Subsequently, a pressure test scheme is developed using the JMeter pressure test tool. Finally, the CPU utilization, response time, and Pod number of PHP-apache applications under the two container elastic scaling strategies are compared and analyzed at the same sampling points. The following sections introduce the load test application, pressure test scheme, and the comparison of the results of the container elastic expansion experiment, respectively.

(1) Load test applications

In order to test the performance of the container elastic scaling strategy, the PHP-apache container application is deployed in the Kubernetes cluster. The PHP-apache container application provides a website access interface that uses HTTP connection on port number 80. To more accurately measure the performance of each container elastic scaling strategy, the configuration information of the load test application PHP-apache is determined, as shown in Table 8.

Table 8 PHP-apache application configuration.

Parameter	Description	
Request CPU	200 m nuclear	
Request interval	0.01 s	
Initial number of copies	1	
Maximum number of copies	10	

(2) Pressure test scheme

After determining the configuration of the load test application, the experimental pressure test scheme utilizes the JMeter tool to manage the workload of the containerized application. The PHP-apache application is tested under two container scaling strategies (HPA and PRL proposed in this article). The stress test scheme ramps up the simulated burst load rapidly by regulating the user traffic and creates jitter load by continuously fluctuating the user traffic in a short time. The specific steps are as follows:

a. Start a thread group with 10 threads to send requests to the application. Each thread should loop 50 times, and the request duration should be 5 min.

b. After 3 min, initiate a thread group consisting of 30 threads to send requests to the application, iterate 50 times, with each request taking 6 min. Concurrently, both thread groups 1 and 2 will be active, causing a sudden surge in access volume to simulate burst load.

c. After 10 min, a thread group containing 10 threads is started to send a request to the application. This process is repeated 50 times, with each request taking 6 min. Once this is completed, thread groups 1 and 2 are finished, and the loop of thread group 3 sends access requests to simulate the jitter load.

The stress testing tool JMeter can simulate a scenario where multiple users simultaneously access the PHP-Apache application interface. The load flow simulated by the JMeter tool lasts for 15 min in total, encompassing the main trends of load changes in the container environment (burst load and jitter load). The simulated load flow is illustrated in Fig. 7.

Figure 7 Simulated workload flow.

(3) Comparison of elastic expansion effect of container

In order to observe the effects of different container scaling strategies on CPU utilization, response time, number of application instances, and SLA violations of the tested applications, the changes in the above performance indicators of PHP-apache applications under simulated load conditions are observed and compared. The actual effects of each container scaling strategy are analyzed based on the trends of performance indicators. Under the simulated load flow, HPA and PRL are realized respectively.

The change in the number of Pods for the two container elastic scaling strategies under burst load is illustrated in Fig. 8. It is evident that at time 5 and time 8, the container elastic scaling strategy PRL proposed in this article initiates an early expansion action, enabling it to adjust to the upcoming peak load promptly and ensuring that the application’s service quality will not be significantly compromised due to high load. Simultaneously, the HPA scaling strategy incrementally increases the number of Pods based on the current load observed. However, due to the startup delay of Pods, creating Pods only when the traffic peak occurs will result in excessively long response times at this stage, impacting the application’s service quality.

Figure 8 Changes of Pod number under burst workload.

The change in the number of Pods for the two container elastic expansion strategies under jitter load is illustrated in Fig. 9. It is evident that at time points 4 and 6, the PRL strategy expands preemptively to handle the upcoming load peak, while the HPA strategy gradually increases the number of Pods during the same period, resulting in a delay. From time point 7 to time point 10, the impact of this strategy mirrors that of the HPA strategy. At time point 11, the PRL strategy initiates contraction in advance, enhancing resource utilization and minimizing waste compared to the HPA strategy. Despite a more significant reduction in Pods at time point 11, the PRL strategy maintains stability at the same Pod level by adjusting subsequent time steps, unlike the HPA strategy. Simultaneously, the HPA scaling strategy decreases the number of Pods gradually based on observed load, leading to significant resource idleness post-traffic reduction and diminishing average resource utilization.

Figure 9 Changes of Pod number under jitter workload.

The CPU utilization changes of the two container elastic scaling strategies under simulated load are shown in Fig. 10. It can be observed that during the load peak at time point 8, the CPU utilization of the PRL and HPA strategies increases less. By anticipating the load peak, Pods are created in advance, ensuring better application service quality during peak loads. At time point 26, the PRL strategy proactively deletes redundant Pods by predicting the low load peak to free up unnecessary resources. In comparison to the HPA strategy, it achieves higher resource utilization during low load stages.

Figure 10 Changes of CPU utilization under simulated workload.

SLA outlines the quality of service requirements between service providers and users. It typically includes criteria such as service stability, reliability, response time, data backup time, and more. Since the container application deployed in this experiment is a web application, the SLA violation threshold is set based on the distribution of response time in the experiment. The strategy ensuring that the average response time does not exceed the violation threshold is considered optimal. The 90% quantile of response time is 146 ms, that is, 90% of the response time does not exceed 146 ms. Therefore, in this experiment, the response time equal to 146 ms is set as the SLA violation threshold.

To further illustrate the impact of the container elastic scaling strategy proposed in this article, two aspects are considered: reducing the response time during the busy period of the application and improving the CPU utilization during the idle period of the application. The RL elastic scaling strategy proposed in Rossi, Nardelli & Cardellini (2019) is used for comparison. The strategy involves elastic scaling, which combines horizontal and vertical scaling actions. A model-based reinforcement learning method is employed to learn the optimal elastic scaling strategy. In order to compare the effectiveness of the results, only the horizontal scaling action is utilized here. The comparison results are presented in Table 9. It can be observed that during burst load occurrences, the PRL elastic scaling strategy initiates early expansion actions, resulting in a lower average response time. In contrast, during jitter load instances, the PRL elastic scaling strategy implements early contraction actions while maintaining higher CPU utilization. This approach ensures that the average response time remains below the SLA violation threshold, thereby preventing service quality issues stemming from excessive contraction. Compared with the RL elastic scaling strategy, PRL utilizes fewer scaling actions during burst loads. This approach can effectively minimize resource wastage, enhance cluster performance and stability. Additionally, it mitigates the impact of scaling operations on applications, preventing performance fluctuations caused by frequent scaling.

Table 9 Performance comparison of three container elastic scaling strategy.

Load type	Mean response time (ms)	Average CPU utilization (%)	Expansion and contraction action (times)	
	Burst load (1–15)	Jitter load (15–30)	Burst load (1–15)	Jitter load (15–30)	Burst load (1–15)	Jitter load (15–30)	
HPA elastic scaling strategy	176.05	83.27	4.26	1.06	6	8	
RL elastic scaling strategy	135.35	129.48	3.53	2.37	6	8	
PRL elastic scaling strategy	130.46	136.94	3.79	2.48	3	8	

Conclusion

This article proposes a method for scheduling edge computing resources based on container elastic scaling. The TE-TCN model is designed to predict the container load. Then, reinforcement learning is used to construct a container scaling strategy, which has two optimization objectives (reducing SLA violation rate and improving resource utilization), and changes the fixed-size action space in traditional reinforcement learning to a time-varying space combined with TE-TCN load prediction value. The method proposed in this article can perform the expansion action faster than the traditional container scaling strategy when the burst load arrives, and reduce the response time of the application. At the same time, the strategy can ensure high CPU utilization in the idle period and avoid waste of resources.

Supplemental Information

Supplemental Information 1 Code.

Supplemental Information 2 Raw data.

Additional Information and Declarations

Competing Interests

Author Contributions

Data Availability

The authors declare that they have no competing interests.

Huaijun Wang conceived and designed the experiments, performed the experiments, performed the computation work, authored or reviewed drafts of the article, and approved the final draft.

Erhao Deng conceived and designed the experiments, performed the experiments, analyzed the data, performed the computation work, prepared figures and/or tables, and approved the final draft.

Junhuai Li conceived and designed the experiments, performed the experiments, performed the computation work, authored or reviewed drafts of the article, and approved the final draft.

Chenfei Zhang conceived and designed the experiments, performed the experiments, analyzed the data, performed the computation work, prepared figures and/or tables, and approved the final draft.

The following information was supplied regarding data availability:

The code and data is available in the Supplemental Files.

The WorldCup98 data is available from: https://ita.ee.lbl.gov/html/contrib/WorldCup.html.

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
