# Peer review of "Edge computing resource scheduling method based on container elastic scaling"

_PeerJ Computer Science, doi:10.7717/peerj-cs.2379_

## Round 0.1 · original submission · Major Revisions

This manuscript proposes an effective edge computing resource scheduling method based on container elastic scaling. Overall, the paper is well-organized and easy to follow. Please revise the manuscript based on the external reviews, including the details of the proposed method, motivation, and presentations.

Reviewer 1 ·

Basic reporting

This paper shows Edge computing resource scheduling method based on container elastic scaling. I have following suggestions

1)The background to resource scheduling and container elastic scaling should be clearly shown. What are the new features to your method? You had better to give Advantages and disadvantages of existing methods, why or form which view you start your research point.

2) In the abstract part, you had better to show your method in step by step in order to show the core work you have done. Form the current abstract, I cannot find the innovation, which is superficial to tell your method. You had better to add the more step details. In introduction part, you have to reconstruct it according to the sequence of background, problem to state,existing methods with disadvantage,your method and from which point to solve. I find the related works are disorder. It should be sorted based on different method or methods. After you give and discuss proposed methods,you should state the problem or summary.

Experimental design

3) The experiment should be improved a lot. Please recheck experiment section that the experiment is hard to read. you have to reconstruct it according to the sequence of experiment target,Data preparation, how to compare?and your metrics. You should give the step of experiment that how to carry out, such as parameter,step, condition in scenario. You should show the configuration, that how to compute? You should reconstruct it according to the sequence of Phenomena, causes, and recommendations. You should give more details and Analysis according to experiment above you carry out.

Validity of the findings

OK for that

Additional comments

none

Cite this review as

Reviewer 2 ·

Basic reporting

1. Adjust Figure 1 to ensure a tight connection between the parts.

2. The font size and initial capitalization of the text in Figure 3 are not standardized, please standardize font size and initial capitalization.

3. The paper should cite up-to-date literature. Some references are no longer aligned with the latest research advances. Please replace them with the most recent literature.

4. In formula 2, the variable $\hat{y}_{t+3}$ should be modified to $\hat{y}_{t+h}$.

Experimental design

1. In the comparison method, ANN, ARIMA, and LSTM have been reported for many years, so they cannot represent the state-of-the-art methods in this field. As a kind suggestion, it is better to replace them with some state-of-the-art methods to validate the performance.

Validity of the findings

1. In the paper, RMSE and MSE are selected as evaluation indexes. It is proposed that the square relationship between RMSE and MSE be used to select one of these indexes. Additionally, the data presented in the paper are problematic. For instance, in Table 4, the MSE of ARIMA and TE-TCN are both 0.0003, yet there is a significant discrepancy in the RMSE between the two, which is an implausible phenomenon. Please verify this issue.

2. In Figure 5, the MSE metrics of the ANN model at 1,5,10 steps are too flat compared to the other models, which is not a normal phenomenon, please check the part.

Additional comments

1. There are several instances in the paper where the abbreviations are repeated and the formatting is not standardized, for example, the full name of MDP plus the abbreviation is repeated in lines 251 and 257, and the formatting in line 251 is Markov Decision Process (MDP), but the formatting in line 257 is Markov Decision Process (MDP). Please check the abbreviations throughout the text to ensure that there are no duplications and to harmonize the formatting.

2. Enlarge the font size of the x-axis and y-axis labels, as well as the legend in Figure 7 to make them clearer.

3. Replacing the color scheme of Figures 8 and 9, consider leaving a small space between the same set of bars.

4. In the results analysis of the abstract, specific values or effects can be displayed.

Cite this review as

Reviewer 3 ·

Basic reporting

The proposed TE-TCN model for load prediction shows promising results compared to other methods like LSTM and ARIMA. The use of reinforcement learning for container scaling strategy is innovative and shows potential for improving resource utilization.

Experimental design

The paper doesn't clearly explain how the TE-TCN model works. For example, the "trend enhancement module" is mentioned but not described in detail.

The paper compares its method primarily with basic approaches like HPA. It should include comparisons with more recent and advanced container scaling techniques.

Validity of the findings

While the WorldCup98 dataset is used, it would be beneficial to see results on more diverse and recent datasets like Google Cluster Data (2019 version) https://github.com/google/cluster-data , Alibaba Cluster Trace (2018): https://github.com/alibaba/clusterdata or Azure Public Dataset (2020): https://github.com/Azure/AzurePublicDataset) ) to demonstrate generalizability.

It would strengthen the paper if a discussion on the practical implications of the improvements in metrics like CPU utilization and response time is added.

Cite this review as

---

## Round 0.2 · accepted · Accept

Based on the authors' response and external reviews, it can be accepted.

Reviewer 1 ·

Basic reporting

I vote to accept this paper

Experimental design

I vote to accept this paper

Validity of the findings

I vote to accept this paper

Additional comments

I vote to accept this paper

Cite this review as

Reviewer 2 ·

Basic reporting

The authors have already fixed the issues and concerns from me since last review.

Experimental design

The authors have already fixed the issues and concerns from me since last review.

Validity of the findings

The authors have already fixed the issues and concerns from me since last review.

Cite this review as